# Mechanisms, Pathophysiology and Currently Proposed Treatments of Chronic Obstructive Pulmonary Disease

**DOI:** 10.3390/ph14100979

**Published:** 2021-09-26

**Authors:** Sarah de Oliveira Rodrigues, Carolina Medina Coeli da Cunha, Giovanna Martins Valladão Soares, Pedro Leme Silva, Adriana Ribeiro Silva, Cassiano Felippe Gonçalves-de-Albuquerque

**Affiliations:** 1Laboratório de Imunofarmacologia, Fundação Oswaldo Cruz (FIOCRUZ), Rio de Janeiro 21040-900, Brazil; sarahrodrigues.bio@gmail.com; 2Laboratório de Imunofarmacologia, Departamento de Bioquímica, Instituto Biomédico, Universidade Federal do Estado do Rio de Janeiro, Rio de Janeiro 20211-010, Brazil; carolina.cunha230@gmail.com (C.M.C.d.C.); giovanna_ciccone@hotmail.com (G.M.V.S.); 3Programa de Pós-Graduação em Ciências e Biotecnologia, Universidade Federal Fluminense, Rio de Janeiro 24020-140, Brazil; 4Laboratório de Investigação Pulmonar, Carlos Chagas Filho, Instituto de Biofísica, Universidade Federal do Rio de Janeiro, Rio de Janeiro 21941-902, Brazil; pedroleme@biof.ufrj.br; 5Programa de Pós-Graduação em Biologia Celular e Molecular, Instituto Oswaldo Cruz (FIOCRUZ), Rio de Janeiro 21040-900, Brazil; 6Programa de Pós-Graduação em Biologia Molecular e Celular, Universidade Federal do Estado do Rio de Janeiro, Rio de Janeiro 20210-010, Brazil

**Keywords:** chronic obstructive pulmonary dysfunction, COPD, pathophysiology, current treatments

## Abstract

Chronic obstructive pulmonary disease (COPD) is one of the leading global causes of morbidity and mortality. A hallmark of COPD is progressive airflow obstruction primarily caused by cigarette smoke (CS). CS exposure causes an imbalance favoring pro- over antioxidants (oxidative stress), leading to transcription factor activation and increased expression of inflammatory mediators and proteases. Different cell types, including macrophages, epithelial cells, neutrophils, and T lymphocytes, contribute to COPD pathophysiology. Alteration in cell functions results in the generation of an oxidative and inflammatory microenvironment, which contributes to disease progression. Current treatments include inhaled corticosteroids and bronchodilator therapy. However, these therapies do not effectively halt disease progression. Due to the complexity of its pathophysiology, and the risk of exacerbating symptoms with existing therapies, other specific and effective treatment options are required. Therapies directly or indirectly targeting the oxidative imbalance may be promising alternatives. This review briefly discusses COPD pathophysiology, and provides an update on the development and clinical testing of novel COPD treatments.

## 1. Introduction

A hallmark of chronic obstructive pulmonary disease (COPD) is the chronic obstruction of the airways. COPD is a progressive condition caused by inhalation of toxic particles or gases [1,2]. Tobacco smoking and inhalation of other pollutants are the leading causes of COPD [3,4,5].

COPD is a major cause of global morbidity and mortality, resulting in increased economic and social burden [1,2,6]. Variance among countries and between different groups in the prevalence of this disease is often directly related to smoking prevalence, although environmental pollution is also a significant risk factor in many countries. The prevalence and burden of COPD will increase in the coming decades due to continued exposure to risk factors and aging of the world population [5,7]. There are many pulmonary and systemic comorbidities in COPD patients, such as bronchiectasis, asthma, heart failure, cardiovascular diseases, sleep apnea, malnutrition, and frailty [8].

The inflammatory process can alter the bronchi, bronchioles, and pulmonary parenchyma, leading to progressive restriction of airflow, resulting in emphysema and chronic bronchitis [9,10]. The pathogenesis of emphysema includes destruction of alveolar septa, increased air space, and loss of elastic recoil due to hyperinflammation and oxidative stress [11,12,13]. Chronic bronchitis involves the overproduction and hypersecretion of mucus by goblet cells, thereby reducing airflow [14] (Figure 1).

## 2. Epidemiology

COPD was estimated to affect 251 million people in 2016, and in 2015, 3.17 million patients worldwide died due to COPD, ranking COPD as the third most deadly disease [15]. The highest prevalence occurs in the Americas [16,17], where its prevalence has increased over the past 20 years. Despite the growing global burden, COPD is neglected in low-income countries, where it is considered a non-communicable disease [18,19]. In Canada, the risk of developing COPD is similar to that of developing diabetes, which is more significant than the risk of developing congestive heart failure [20]. 

According to the Centers for Disease Control and Prevention, the United States had 153,445 deaths due to COPD in 2019. In 2018, 5.1% of adults (approximately 12.8 million people) were diagnosed with COPD [21,22]. COPD prevalence is higher in women than in men, increasing exponentially with age. Race/ethnicity and socioeconomic status are also risk factors [23,24].

In 2010, most COPD-related deaths occurred in low- and middle-income countries. No population-based epidemiological studies have been conducted in these countries [25]. In developing countries, exposure to biomass can be a risk factor for non-smoking related COPD, impacting strategies for prevention and treatment [26]. Studies have also shown evidence supporting a relationship between air pollution and COPD [27]. Hence, COPD prevalence is generally higher than health authorities’ estimates, rendering it an underdiagnosed disease. There are several reasons for this underestimation, including lack of robust diagnostic standards, variation in lung function tests, inconsistent use of COPD terminology, and limited government funding [25,28,29]. The disease COVID-19 caused by the Severe Acute Respiratory Syndrome Coronavirus 2 (SARS-CoV-2) was declared pandemic by the WHO in March 2020, and since then, it has continued to spread, mainly in elderly people and people with comorbidities. Diabetes, obesity, cardiovascular diseases, and respiratory diseases are among the comorbidities linked to increased severity in cases of COVID-19 [30]. Data relating COPD and COVID-19 are contradictory, with the incidence of COVID-19 in COPD patients being lower than expected. The reason for this is unclear [31]. However, it is essential to highlight that patients with COPD are at increased risk of developing severe COVID-19 [32].

## 3. Pathophysiology

In normal alveolar septa, elastic fibers located subepithelial layer are predominant, which confer resistance to connective tissue, allowing deformability and passive recoil without energy input [33]. Elastic fibers are mechanically connected to collagen fibers via microfibrils and/or proteoglycans [34,35]. Traditionally, elastic fibers are responsible for lung elasticity within a normal lung volume range, while collagen fibers are responsible to halt lung volume when it approaches the total lung capacity [35]. 

The breakdown of elastic fibers, so-called elastolysis, is one of the hallmarks of emphysema, an important phenotype contributing to COPD [36]. COPD is characterized by progressive airflow limitation that is not fully reversible, associated with an abnormal inflammatory response of the lungs to noxious particles or gases [2,37]. Emphysema is the result of destruction of alveolar walls, which leads to reduced gas exchange, permanent airspace enlargement, loss of elastic recoil, hyperinflation, and expiratory flow limitation [38,39,40]. As a consequence of fiber destruction by metalloproteinases, there are changes in collagen- and elastic-fiber organization [11]. These features affect the lung’s tissue stability and mechanical properties, contributing to lung function decline overtime and accelerating disease progression [41,42].

### 3.1. Diaphragm Dysfunction and COPD

The dynamics hyperinflation results in diaphragm mechanical disadvantage leading to dysfunction [43,44]. Clinical studies using ultrasonography in in-hospital patients have shown its ability to detect diaphragm weakness, resulting in increased hospital length of stay [45,46]. The diaphragm weakness acquired during exacerbation can be explained by: (1) elevated number of inflammatory cells in the lungs [47,48]; (2) oxidative stress and damage within the diaphragm [49]; (3) diaphragm remodeling [44]; (4) maintenance of hyperinflated areas, which jeopardize diaphragm performance [50,51]; and (5) changes in mitochondrial dynamics. Several studies have shown that mitochondria are dynamic organelles with the ability to change morphology and function according to the pathologic situations through fusion and fission processes [52]. Mitochondrial fusion is mediated by proteins located at the external mitochondria membrane, such as mitofusin 1 (MFN1), mitofusin 2 (MFN2), and optic protein factor 1 (OPA 1). These proteins hydrolases GTP and promote mitochondria fusion, which allows DNA, protein, and metabolites sharing. Mitofusins act toward the external membrane forming homo- and heterodimers [53,54], while OPA1 acts toward the internal membrane. Thus, the loss of these proteins may lead to mitochondrial DNA damage [55], affecting bioenergetics function [56]. The mitochondrial fission is characterized by mitochondria fragmentation, and the main objectives are: (1) to increase the mitochondria numbers to distribute to new cells during mitosis [57]; (2) to transport to other regions of the cell; and (3) to signalize injured cells and forward them to mitophagy [58] and apoptosis. First, fission occurs through the inhibition of mitochondrial fusion protein. Second, the fission process demands the presence of mitochondrial fission, such as dynamin-related protein 1 (DRP1) [59], which interacts with human fission factor (Fis 1) and mitochondria fission factor (MFF). Thus, the extensive activation of DRP1 may increase mitochondrial fragmentation, increasing reactive oxygen species followed by decreased ATP production [60,61].

### 3.2. Pulmonary Arterial Hypertension and COPD

Pulmonary arterial hypertension (PAH) and diaphragm dysfunction are commonly observed during COPD progression, contributing to exacerbations [62,63]. Exacerbations are acute episodes caused by viral and bacterial infections, which worsen airway inflammation, cause lung function decline followed by hospitalization, and increase mortality [64,65]. COPD patients who present PAH have decreased survival rate compared to COPD patients at similar severity but without PAH [62]. One probable explanation is that PAH is associated with vascular remodeling, likely due to collagen fibers accumulation beneath pulmonary vessels, which leads to vessel narrowing overloading the right ventricle [66,67,68]. Furthermore, during PAH associated with COPD development, there is an increase in the pulmonary inflammatory process and the release of vasoactive agents, such as thromboxane A2. It can induce vascular constriction and further increase vascular resistance [69].

### 3.3. Reactive Oxygen Species and COPD

In addition to inflammation, COPD is characterized by an imbalance between proteases and their inhibitors, oxidative stress, and infections that generate disease symptoms [70,71]. Prognosis for COPD patients depends on different factors, including disease severity, body mass index, and age [10]. Patients display increased numbers of neutrophils, macrophages, and T cells in the lungs, increasing chemotactic mediators [71]. In addition to pulmonary inflammation, there is also systemic inflammation with increased levels of fibrinogen, C-reactive protein (CRP), serum amyloid A (SAA), and pro-inflammatory cytokines tumor necrosis factor-alpha (TNF-α), interleukin-6 (IL-6), and IL-8 in the serum [72]. Cigarette smoke (CS) is the primary source of oxidant agents in the lungs, but inflammatory cells and phagocytes residing in the respiratory tract also generate reactive oxygen species (ROS) in the lungs [70,73]. Increased nicotinamide adenine dinucleotide phosphate (NADPH) activity in epithelial cells, phagocytes, and myeloperoxidase in neutrophils is responsible for ROS production in patients with COPD [73]. Oxidative stress generated by CS results in nuclear kappa B (NF-κB) activation, producing inflammatory mediators that foster tissue damage [70,73]. NF-κB activation induces cytokines, chemokines, and cell adhesion molecules, which are boosted by bacterial or viral infections, exacerbating disease symptoms [74]. Oxidative stress is the primary cause of COPD pathogenesis, triggering apoptosis, extracellular matrix remodeling, inactivation of protease inhibitors, mucus secretion, NF-κB activation, mitogen-activated protein kinase (MAPK) activation, chromatin remodeling, and pro-inflammatory gene transcription [71,74,75]. 

Healthy lungs possess enzymatic and non-enzymatic antioxidant mechanisms that counteract oxidative stress. Non-enzymatic mechanisms involve glutathione (GSH), vitamin C, uric acid, vitamin E, and albumin. Enzymatic mechanisms rely on superoxide dismutase (SOD), catalase, and glutathione peroxidase (Gpx) [72]. Exposure to CS decreases intracellular GSH levels, boosting oxidative stress in COPD patients [76]. The transcription factor nuclear factor erythroid 2-related factor 2 (Nrf2) is crucial for regulating the cellular antioxidant response and preventing ROS-induced injury. Nrf2 regulates the expression of genes encoding enzymes that regulate oxidative stress, including the typical phase 2 detoxifying enzyme hemoxygenase-1 (HO-1). Decreased Nrf2 pathway stimulation in peripheral lung tissue and alveolar macrophages is associated with increased susceptibility to and severity of COPD [77].

In the next section, we discuss both immune and structural cells in COPD pathophysiology.

## 4. Inflammatory Cells and Mediators

The terminal bronchioles and lung parenchyma are the main regions affected by COPD inflammation, and are characterized by infiltrating macrophages and CD8^+^ T-cells. Macrophages are primarily present within the lungs, while CD8^+^ T-cells cause alveolar epithelial cell apoptosis and destruction through release of perforins and TNF-α [9]. Macrophages and neutrophils are involved in ROS generation during COPD [70]. In response to macrophages and neutrophils, alveolar epithelial cells release leukotriene B4 (LTB4), a chemotactic factor that attracts immune cells [9,78]. Macrophages and pulmonary cells also produce IL-8/CXCL8 [79] and growth-related oncogene (GROα)/CXCL1, which amplify the inflammatory response by attracting more leukocytes from the blood to the inflammatory site [80]. Patients with COPD display smoking-linked ICAM-1 augmentation in epithelial cells. ICAM-1 is an adhesion molecule that is crucial for leukocyte migration. It is highly expressed in patients with severely limited airflow, and is associated with increased risk of viral and bacterial infections [81].

The imbalance between proteases and their inhibitors plays a crucial role in COPD pathogenesis. Proteases, including neutrophil elastase (NE) and proteinase 3, degrade connective tissue components, especially elastin, leading to emphysema [78,82]. Elastin is detected in the serum of patients with COPD due to massive tissue destruction [82]. α1-antitrypsin can inhibit NE, but it is reductively inactivated in COPD patients [83]. Elastic fiber damage may lead to collagen deposition in the pulmonary parenchyma, leading to alveolar septa destruction and alveolar distention [3]. Metalloproteinases (MMPs) attack the extracellular matrix, causing the release of elastin fragments that attract monocytes to the lungs. MMPs are also involved in recruiting pulmonary macrophages, thus raising proteolytic and inflammatory activity, thereby playing an essential role in COPD progression [11].

NE regulates expression of the *MUC5AC* gene, which encodes the gel-forming mucin of the respiratory tract, through an ROS-dependent mechanism [70] associated with airway obstruction and disease severity [84,85,86,87]. Bacterial components, such as lipopolysaccharide (LPS), and cytokines, such as IL-9, TNF-α, and IL-1β, enhance *MUC5AC* gene expression [70,79], amplifying the inflammatory process. 

Therefore, ROS, inflammatory mediators, and proteolytic enzyme production can initiate, enhance, and aggravate tissue damage, and exacerbate lung injury, resulting in COPD development and progression (Figure 2).

### 4.1. Alveolar Epithelial Cells

Alveolar epithelial cells serve as a mechanical barrier to harmful stimuli [88]. Exposure of these cells to CS and other pollutants activates several intracellular signaling pathways, which induce pro-inflammatory mediators, including CXCL8/IL-8, GM-CSF, ICAM-1, and TNF-α, regulating the influx of inflammatory cells [89]. 

Vascular endothelial growth factor (VEGF) and hepatocyte growth factor (HGF) are vital for maintaining alveolar epithelial cell integrity [90]. Low VEGF and HGF levels have been associated with alveolar epithelial cell apoptosis in COPD patients [91]. After exposure to inhaled irritants, small airway epithelial cells increase TGF-β expression, which stimulates fibroblasts to differentiate into myofibroblasts that produce extracellular matrix (ECM), leading to local fibrosis [92]. While MMP-9 is released by cells involved in immune defense, such as macrophages, MMP-2 is synthesized by fibroblasts, and has been associated with chronic tissue remodeling, leading to abnormal tissue changes [93]. Furthermore, intense ROS production can disrupt surfactant secretion by alveolar epithelial cells, leading to alveolar collapse and high airway resistance due to interdependence, a typical sign of emphysema [89,94].

ROS production is also associated with mitochondrial DNA damage, causing mitochondrial dysfunction [95]. Such dysfunction has been reported in airway smooth muscle cells in COPD patients [96]. CS inhibits mitochondrial respiratory function, reducing ATP production, which induces mitophagy in alveolar epithelial cells [97]. Mitochondrial ROS generation has been associated with surfactant changes, thus interfering with alveolar epithelial cell stability [98].

Inhaled toxic agents damage alveolar epithelial cells, causing the release of damage-associated molecular patterns (DAMPs), observed in bronchoalveolar lavage fluid (BALF) from COPD patients [99]. CS induces alterations in alveolar epithelial cells, leading to alveolar-capillary barrier dysfunction [100]. Damage to alveolar capillaries facilitates pathogen entrance, increasing the risk of exacerbating symptoms [101]. 

### 4.2. Goblet Cells

Goblet cells are essential to the immune system because of their ability to secrete mucus, antioxidants, protease inhibitors, and defensins, to maintain the epithelial barrier against infectious agents [90]. The epidermal growth factor receptor (EGFR) induces cell proliferation by promoting production of transforming growth factor-alpha (TGF-α), resulting in mucus production [102,103]. EGFR activation is increased in COPD patients [104], elevating their cell proliferation and mucus production [14]. COPD patients display high production of intracellular mucin, mainly MUC5AC [105], and high NF-κB activation and cytokine release [106]. CS and ROS also amplified MUC5AC expression via EGFR activation. MUC5AC production leads to respiratory tract hypertrophy and hyperplasia [70], facilitating disease progression [107]. Mucus hypersecretion also increases viscosity, decreases antibacterial molecule production, aggravates airflow restrictions, and increases the risk of lung infections [108].

### 4.3. Alveolar Macrophages

Alveolar macrophages play a crucial role in innate and adaptive immune responses. They are involved in capturing and processing inhaled harmful agents, and stimulating the immune response by releasing inflammatory mediators, including TNF-α, CXCL1, CXCL8, CXCL9, CXCL10, CCL2, leukotriene B4 (LTB4), and ROS [90]. These mediators also recruit monocytes, neutrophils, and lymphocytes to the inflammatory site. Macrophages also secrete elastolytic enzymes, including MMPs and cathepsins [109]. Similar to neutrophils, macrophages generate ROS through NADPH oxidase (NOX) when stimulated by CS oxidants [110]. In functional NOX2-deficient mice, decreased ROS production provided protection against emphysema [111].

Macrophages can be polarized into either M1 or M2. M1 macrophages display classic activation and have a more inflammatory profile, secreting pro-inflammatory cytokines [112]. They are potent effector cells that specialize in killing microorganisms [113]. M2 macrophages are considered less inflammatory, due to their release of anti-inflammatory cytokines, such as IL-10, and participate in tissue remodeling and repair [114]. In COPD, type M1 is more prevalent than M2, indicating a more accentuated inflammatory response [115], denoted by the release of pro-inflammatory mediators including CCL2 and CXCL1, which enhance cell recruitment, and explaining the higher number of macrophages found in the pulmonary parenchyma, BALF, and sputum of COPD patients [90,116]. Despite their high numbers, macrophages in COPD have low phagocytic and efferocytotic abilities [117]. This altered behavior is associated with exogenous ROS-induced oxidative stress, which is also responsible for altering mitochondrial function, resulting in uncontrolled ROS production [118]. An inadequate macrophage response favors bacterial infection, with increased risk of developing pneumonia and exacerbated symptoms. Therefore, the decrease in phagocytic and efferocytotic functions is related to COPD progression [119]. Macrophage dysfunction is linked to oxidative stress caused by ROS [120]. In healthy individuals, upon oxidative stress, Nrf2 activates a cellular antioxidant response, but Nrf2 activation in macrophages of COPD patients is attenuated [77]. 

The increase in elastolytic enzymes, such as MMPs, induces ECM degradation and alveolar wall destruction [121]. Alveolar macrophages are pivotal to COPD development, because even after smoking cessation, their dysfunction may continue to contribute to disease progression [122]. These cells play a primary role in the pathology of COPD, and the integrity of their functions is directly related to the degree of inflammation and disease severity.

### 4.4. Lymphocytes

Lymphocytes can cause alveolar destruction in patients with COPD [123]. Lymphocyte-activating IL-17 and IL-22 levels are increased in patients with COPD [124]. CD8^+^ cells produce pro-inflammatory cytokines, including IL-2, gamma interferon (IFNγ), and TNFα, and chemokines, including CXCL10 and CCL5, which recruit other inflammatory cells [125]. All these mediators increase in COPD patients [126], taking part in COPD pathogenesis and autoimmune response [127]. CD8^+^ cells release perforins, granzyme B, and TNF-α, causing cytolysis and apoptosis of alveolar epithelial cells, resulting in emphysema development [128].

CD4/CD28^null^ T cells are a pro-inflammatory subset of T helper lymphocytes, whose main characteristic is loss of CD28, a necessary co-stimulatory receptor for CD4+ T cell activation, proliferation, and survival [129]. The increase in these T cells in COPD patients is linked to impaired lung function. Chronic exposure to CS causes a reduction in the costimulatory molecule CD28, increasing expression of perforin, granzyme B, and receptors in NK cells and T cells [130]. COPD patients are resistant to immunosuppression induced by corticosteroids compared to control individuals with higher counts of CD8/CD28 T cells [131]. T cell senescence in COPD patients may be associated with reduced histone deacetylase 2 (HDAC2) expression in CD8/CD28^null^ T cells [132]. T cell receptor downregulation leads to an inadequate response to infection, causing either autoimmune disease [133] or increased susceptibility to COPD complications [125].

Regulatory T cells (Tregs) cause immune suppression at the inflammatory site, producing IL-10 and TGF-β1 [134]. FOXP3 is considered a specific marker of Treg cells [135]. In COPD patients, increased levels of Tregs have a pronounced effect, disrupting effector T cell response and tolerance to self-antigens [136].

Patients with severe COPD have a higher number of B cells in their small airways [137]. They also have higher levels of B cell activation factors in lymphoid follicles [138], and B cells secrete autoantibodies against carbonylated proteins, which form due to oxidative stress [139]. Antibodies against elastin, observed in patients with emphysema, were the first evidence of a relationship between autoimmunity and COPD [140]. COPD patients also present with high levels of anti-endothelial and anti-epithelial antibodies [141]. Similar to rheumatoid arthritis patients, they have citrullinated proteins in the lungs, which can induce autoantibodies [142]. Furthermore, COPD patients display a high percentage of apoptotic cells in follicles, suggesting immune dysfunction [143].

Interestingly, the increased number of neutrophils and B lymphocytes correlates with disease severity [144,145,146]. Lymphocytes play an essential role in the autoimmune effects in COPD patients. The ratio of neutrophils to lymphocytes has been suggested for use as a prognostic marker for predicting exacerbations in COPD patients [147,148]

### 4.5. Neutrophils

COPD patients have increased neutrophil numbers in sputum and BALF [149]. Smoking stimulates the release of granulocytes from the bone marrow and their survival in the respiratory tract [150]. Chemotactic factors, including LTB4, CXCL1, CXCL5, and CXCL8 derived from alveolar macrophages, epithelial cells, and T cells, can boost neutrophil migration in COPD patients [90].

Expression of adhesion molecules such as E-selectin is increased in COPD patients. E-selectin is expressed on endothelial cells and is critical for neutrophil recruitment [151]. Likewise, myeloperoxidase (MPO) and human neutrophilic lipocalin (HNL) are high in the airways of COPD patients [152]. Upon arrival into the lung inflammation site, activated neutrophils secrete serine proteases (including neutrophil elastase (NE)), cathepsin G, protein-3, MMP-8, and MMP-9, causing alveolar damage [90,153]. Neutrophils also produce neutrophil extracellular traps (NETs), increasing lung tissue damage in COPD patients [154,155].

Although neutrophils may secrete ROS as a defense mechanism in COPD patients, ROS production exceeds physiological levels [156,157]. Excess ROS can alter neutrophil migratory patterns [158], activate granular proteases, induce NET formation [159], and inactivate alpha-1-antitrypsin (α1AT), which in turn promotes inflammation [160]. 

Neutrophil gelatinase-associated lipocalin (NGAL) has been suggested as a systemic marker for COPD [161] because its levels are high in induced sputum and bronchiolar lavage fluid from COPD patients [162]. NGAL is secreted by neutrophils and other cells and possesses antimicrobial properties that can reduce bacterial growth. However, when bound to MMP-9, NGAL extends MMP9 enzyme activity, promoting tissue destruction [163]. Granulocyte-colony-stimulating factor (G-CSF) stimulates neutrophil production in the bone marrow, and promotes their survival, priming, and function. Neutralization, inactivation, or blocking of G-CSF causes inflammation and tissue damage, controls monocyte influx into the lungs, initiates neutrophil apoptosis, and mitigates COPD symptoms [164,165]. Neutrophil-mediated inflammation is critical for COPD development and progression [166]. Studies investigating neutrophil infiltration and activity may shed light on the role of these cells in COPD pathogenesis.

## 5. Genetic and Epigenetic Regulation

CS and oxidative stress cause alterations in histones, including acetylation/deacetylation and methylation/demethylation patterns, resulting in DNA damage, cellular senescence, and pulmonary cell apoptosis, in addition to pro-inflammatory gene expression [167]. Studies have shown that DNA double-strand breaks (DSBs) are among the most lethal forms of DNA damage caused by smoking and oxidative stress. If not repaired, they cause cellular senescence and apoptosis [168]. Oxidant enzyme encoding genes, including cytochrome P450 family 2 subfamily C member 18 (*CYP2C18)* and aryl hydrocarbon receptor nuclear translocator-like 2 (*ARNTL2*), are upregulated in COPD. Other antioxidant genes may undergo mutations, including polymorphisms in glutathione S-transferase (*GST*) *M*1, glutathione S-transferase pi 1 (*GSTP*1), superoxide dismutase 3 (*SOD*3), and epoxide hydrolase 1 (*EPHX*1), being related to lowering lung function and COPD severity [70]. Furthermore, the correlation between epigenetics and the production of inflammatory cytokines may also be linked to disease progression [169]. 

HDACs play a vital role in regulating the inflammatory response. HDACs downregulate oxidative stress sensitive inflammatory gene expression [170]. Macrophage accumulation in the lungs of COPD patients increases the secretion of inflammatory mediators and elastolytic enzymes due to NF-κB activation [171] and the reduction of HDAC2 activity [172]. HDAC2 activity is reduced in alveolar macrophages and lung tissue in COPD patients [77], and this reduced activity is linked to increased histone acetylation at the IL-8 promoter (FISCHER; VOYNOW; GHIO, 2015). CS, oxidative stress, nitrative stress, and aldehyde reduce HDAC2 expression in the lungs [167,173]. Decreased HDAC2 in the lungs further impairs Nrf2 activation, decreasing its half-life, impairing its orchestrated antioxidant defense [70,174], and increasing NF-κB RelA/p65 subunit activation, and thus, increasing the transcription of pro-inflammatory genes [167].

Mucus hypersecretion observed in COPD patients involves epigenetic mechanisms as DNA methylation and histone modification. COPD downregulates HDAC2, causing upregulation of the MUC5AC gene, leading to mucin production and mucus hypersecretion. Conversely, upregulation or increased HDAC2 activity can downregulate MUC5A, diminishing mucus secretion [175].

CS exposure also dysregulates in the expression of small non-coding RNA microRNA (miRNA). Izzotti et al. reported the first evidence of miRNA expression alterations caused by CS, namely downregulation of 24 miRNA involved in apoptosis, proliferation, and angiogenesis in lung [176]. The analysis of the miRNA pattern is important because miRNA senses the environmental stresses, causes phenotype changes in a cell- and tissue-specific way, being potentially used in prognostics, and contributes to the COPD pathogenesis [177,178].

Advances in studying miRNA-based treatment in COPD are promising. Corticosteroids function partially through epigenetic mechanisms as miRNAs. miR-708 and miR-155 were downregulated and miR-320d and miR339-3p can be upregulated by corticoids. miR320d-increased expression diminished the activation of NF-kB signaling. miRNAs affected by corticosteroid treatment in patients with moderate to severe COPD can be considered therapeutic targets in COPD. The miR-223 plays the opposite role. miR-223 directly targets HDAC2 because miR-223 overexpression represses the activity of total HDAC and HDAC2 in pulmonary endothelial cells. COPD population has an inverse correlation between HDAC2 and miR-223 levels. The increase of HDAC2 could diminish the insensitivity GC. This miR-223 can decrease treatment efficacy in COPD patients. Several miRNAs have been modified in COPD and by classical COPD treatments. The identification of these miRNAs and description of their roles through their up- or downregulation could contribute to treatment in the future [179].

Extracellular vesicles carrying extracellular miRNA can be used to diagnose and treat COPD because miRNAs can be delivered in the specific site of action. Exosomal miRNA can be considered biomarkers for diagnosis or prognosis for COPD. A specific miRNA that is important to the better disease outcome can be delivered to the disease site through extracellular vesicles [180].

## 6. Treatments and New Therapeutic Approaches

Although new drug searches target different mechanisms, most drug candidates fail to reach the clinical stage of development, or fail in this phase. Therefore, management of COPD still depends on the use of bronchodilators and corticosteroids [181]. Airflow limitation occurs due to loss of elastic recoil and augmented airway resistance. Despite airflow limitation being the hallmark COPD characteristic, the primary symptom is dyspnea [182], which is related to increased resistive work. Disease progression and dynamic lung hyperinflation progressively increases residual volume after expiration, complicating the inspiratory process [183,184]. Bronchodilators relieve dyspnea by reducing resistive work and airway resistance [183,185]. Spirometry provides a more global assessment of airflow limitation, while computed tomography allows visualization of the anatomical location of the disease, enabling morphological characterization and quantitative analysis of severity contributing to phenotyping [186]. The focus on specific individual characteristics has stimulated research into treatments targeting fundamental disease mechanisms [169]. There are no specific effective pharmacological treatments for emphysema, except for those targeting α1 antitrypsin deficiency [187]. Modes of therapy administration include self-infusion, aerosol, and subcutaneous administration. Gene and recombinant therapies are under development [188,189], and intravenous therapy using α1 antitrypsin derived from human donor plasma has proven to be safe [187].

Corticosteroids (one of the main treatments used in COPD), delivered by oral administration or inhalation, are highly effective anti-inflammatory drugs for asthma. Gene transrepression caused by corticosteroids decreases NF-κB activity [74,190]. Nevertheless, corticoid treatment displays no anti-inflammatory effects in COPD patients. Corticosteroid resistance is primarily caused by inactivation of HDAC2, which is essential for glucocorticoid receptor (GR) repressor activity, which mediates the anti-inflammatory effect of corticosteroids [74,167,190]. HDAC activity represses several activated inflammatory genes, thereby inhibiting oxidative stress [191]. Reduced HDAC2 activity is observed in COPD patients [192,193]. Restoration of HDAC2 and Nrf2 levels overcomes corticosteroid resistance in COPD. Inhibition of the PI3K/Akt/p70S6K signaling pathway restores nuclear HDAC2 expression and activity. Increasing nuclear Nrf2 levels also enhances HDAC2 levels, indicating HDAC2 and Nrf2 involvement in restoring corticoid sensitivity in COPD [194].

Antioxidants and nitric oxide synthesis inhibitors can restore corticosteroid sensitivity in COPD [173]. Corticosteroids associated with bronchodilator therapy are used to prevent exacerbations [20,195]. Bronchodilators, such as long-acting β2 agonists (LABA) and long-acting muscarinic antagonists (LAMA), have beneficial effects against airflow limitation and exercise intolerance. During COPD exacerbations, LAMA is better than LABA, even when LABA is associated with an inhaled corticosteroid [20]. Unfortunately, chronic corticosteroid use can increase the risk of pneumonia in patients with severe COPD, advanced age, comorbidities, such as cardiovascular disease, skeletal muscle wasting, lung cancer, and osteoporosis, or a history of recently diagnosed pneumonia [196,197], because they are immunomodulators and immunosuppressors.

COPD exacerbations are linked to oxidative stress, promoting changes in signaling by pro-inflammatory kinases and transcription factors, steroid resistance, extracellular matrix remodeling, and mucus hypersecretion [70,198,199]. COPD exacerbations are linked to oxidative stress since oxidative stress impairs responses against pathogens and, consequently, contributes to exacerbations induced by viruses and bacteria, causing even more airway inflammation and more exacerbation, thus forming a repeated cycle [200,201]. Thus, therapies targeting oxidative imbalance are promising alternative COPD treatments [70,198,199]. Natural or synthetic antioxidants ameliorate COPD. A large number of molecules act as antioxidants, including thiol compounds (for example, GSH, N-acetyl-L-cysteine, (NAC) [202], N-acystelyn, (NAL) [203], erdosteine [204], fudosteine [205]), polyphenolic compounds derived from the diet (e.g., curcumin [206,207], resveratrol [208], lycopene [209], alpha-lipoic acid [210], and apocynin [198,211]), Nrf2 activators (e.g., CDDO-imidazolide and sulforaphane), antioxidant vitamins (e.g., vitamins C and E) [212], iNOS inhibitors [213], lipid peroxidation inhibitors/blockers, lazaroids/tirilazad [214], myeloperoxidase inhibitors, specialized pro-resolving lipid mediators [198], omega-3 fatty acids [215], and vitamin D [198]. Antioxidants act by decreasing free radical levels, and inflammatory gene expression [71,200,216]. Diet plays a central role in protecting against airway diseases. Carotenoids, vitamin D, vitamin E, vitamin C, curcumin, choline, and omega-3 fatty acids help protect against asthma, COPD, and lung cancer [217,218]. 

COPD treatment using anti-inflammatory compounds remains a challenge due to the complexity of inflammation and related comorbidities. Bronchodilators can reduce inflammation, but they can only be used for a short time, and are not effective in COPD patients [20]. Currently, no therapies effectively reverse COPD pathology. Minimizing COPD progression is an alternative therapeutic strategy. Decreasing oxidative stress and inflammation can improve quality of life and increase survival [72]. Supplementation, therapeutic administration, and/or the use of multiple antioxidants may benefit COPD patients by increasing endogenous antioxidant levels [74,219]. 

### Role of Medications of Each Drug in Patients with COPD

Bronchodilator drugs are currently used to treat patients with COPD to improve symptoms of the disease. β2-adrenergic receptors are present in the bronchi smooth muscles and are G protein-coupled in the cell membrane; when stimulated, they increase the activity of adenyl cyclase, an enzyme that catalyzes the conversion of ATP to cyclic adenosine monophosphate (cAMP). cAMP inhibits intracellular calcium release, decreasing the influx of calcium through the membrane, relaxing smooth muscles, and dilating the airways. Adrenergic receptor agonists may be short-acting bronchodilators (SABA), such as albuterol, used for rapid relief of acute symptoms, or long-acting bronchodilators (LABA), such as formoterol, indacaterol, salmeterol, and tiotropium, used to relieve the most common and persistent symptoms, such as cough and dyspnea [216,220]. Muscarinic and anticholinergic antagonists are short-acting (SAMA), such as levalbuterol, or long-acting (LAMA), such as glycopyrrolate, umeclidinium, arformoterol, and revefenacin. These drugs regulate bronchomotor tonus by stimulating their bronchi muscles-specific receptors. These receptors are G protein-coupled and have five subtypes, among them M1 and M3, which are the primary drugs targets for presenting effects of improving bronchoconstriction and mucus secretion, resulting in improved lung function and dyspnea [221,222].

Corticosteroids can suppress mucus production and decrease airway obstruction due to the suppression of mRNA expression of proteins encoding the MUC5AC gene [222]. Prednisolone and budesonide are used for COPD treatment in combination therapy with bronchodilators and improved symptoms in patients with a history of multiple severe exacerbations [223]. The activation of β2-adrenergic receptors potentiates the anti-inflammatory effect of corticosteroids by increasing the glucocorticosteroid receptor translocation from the cytoplasm to the nucleus [224,225]. Combining corticosteroid therapy with bronchodilators or double-acting bronchodilators (muscarinic antagonists and β2-adrenergic agonists—MABA) has proven beneficial to treat COPD patients’ symptoms and exacerbation [225,226,227]. Viral and bacterial infections are the most frequent cause of exacerbation, which include *Haemophilus influenzae* (NTHi), *Moraxella catarrhalis*, *Streptococcus pneumoniae*, *Pseudomonas aeruginosa*, human Rhinovirus (HRV), Influenza virus, Coronavirus, and Respiratory syncytial virus (RSV) [65]. Antibiotics are being tested to treat bacterial origin exacerbation, such as aismigen, levofloxacin, and ciprofloxacin, already used in tuberculosis and sinusitis treatment. Azithromycin is being tested in viral exacerbations in a mechanism involving interferon response and decreased inflammatory mediator production. Table 1 shows updated research on new drugs for COPD treatment. The tables include drug names, administration form, drug target, registration study number, phase of development and current status, and whether the drug described has already been approved and used or not for the treatment of COPD or other disease COPD comorbidities, as exacerbation because of viral and bacterial infections is the leading cause of hospitalizations and worsening of symptoms. Table 2 shows the current studies treating exacerbations due to bacterial and viral infection. Table 3 shows the different methodologies used in pre-clinical studies. The methodology shows how studies of current clinical treatments for COPD were caried out. The results of these studies are shown in the tables below. We will discuss some of these next. 

Atorvastatin and simvastatin decrease cytokine and leukocyte levels, reduce oxidative stress markers, and improve lung repair [228,229]. Statins can modulate the lung’s extracellular matrix composition because statins may directly regulate MMPs or their biological inhibitors, the TIMPS (inhibitors of matrix metalloproteinases), improving lung function through structural changes [230]. 

Curcumin obtained from turmeric *(Curcumina longa)* is a polyphenol with antioxidant and anti-inflammatory properties [206,207] that modulates glutathione levels and inhibits IL-8 release in lung cells [198]. Studies have shown that curcumin treatment inhibits the increases in neutrophils and macrophages in the BALF of mice exposed to CS and attenuates increases in air space in mice exposed to CS or porcine pancreatic elastase [207].

Eucalyptol (1,8-cineole) is a promising adjunct or anti-inflammatory therapy for COPD and exacerbations [231,232,233] that promotes bacterial elimination in lungs exposed to tobacco, reducing damage to ciliated cells and suppressing expression of MUC5AC in the lungs [234]. Eucalyptol promoted pulmonary repair and decreased levels of MPO, TNF-α, IL-1β, IL-6, KC, TGF-β1, and neutrophil elastase [235]. 

LJ-2698, an adenosine A3 receptor antagonist, significantly attenuated increases in air space, improved lung function, inhibited matrix metalloproteinase activity and lung cell apoptosis, induced increases in anti-inflammatory cytokines produced by macrophages, and significantly increased the number of M2 macrophages [236]. LJ-529, a partial peroxisome proliferator-activated gamma receptor (PPARγ) agonist, showed similar results, and induced expression of PPARγ target genes, which play a role in regulating inflammation [237].

Therapy with mesenchymal stromal cells (MSCs) is a thoughtful approach to treating pulmonary diseases, including COPD, mainly based on the immunosuppressive role of MSCs. MSCs are promising adjuvants, used in combination with other treatments, that can improve pulmonary function and decrease inflammation through their anti-inflammatory, antioxidant, microbicidal, and angiogenic action [238,239]. 

Thioredoxin (Trx) is an essential regulator of the body’s redox balance, which can benefit COPD patients through varied mechanisms of action, either as a primary treatment or as a coadjuvant with other treatments [240,241]. Trx also improves resistance to corticosteroids [240], inhibits elastase-induced emphysema [242], decreases neutrophilic inflation [243], and blocks the production of inflammatory cytokines [241]. Further clinical studies are required to verify its effectiveness in COPD treatment [240].

New treatment research focuses on LABA [244] and LAMA [245], such as vilanterol and umeclidinium, including inhibitors of inflammatory mediators, such as canakinumab [246], infliximab [247], and mepolizumab [248]. Phosphodiesterase (PDE) inhibitors (roflumilast and the M3 receptor antagonist glycopyrrolate) [249] exert anti-inflammatory and bronchodilator effects by inhibiting an enzyme involved in the degradation of second messengers [250]. Nevertheless, few clinical trials are currently assessing decreases in oxidative stress, which is a significant factor in COPD and its comorbidities. 

Figure 3 shows the main mechanisms of action of drugs under development for COPD treatment.

## 7. COPD and COVID-19

SARS-CoV-2 has affected more than 150 million people worldwide, and has caused more than 3 million deaths [251]. Patients with pulmonary comorbidities, such as COPD, belong to the high-risk group [252,253]. High-risk group patients are more likely to develop COVID-19 with worse progression and prognosis [253,254]. They present a four times greater risk of developing the severe form of the disease [254], and smokers have a higher risk of severe complications and a higher mortality rate [253]

SARS-CoV-2 uses the ACE-2 receptor (angiotensin-converting enzyme 2) to enter cells [255,256]. Increased airway expression of the ACE-2 receptor in COPD patients and smokers correlates with their increased risk of developing COVID-19 [252,257]. COPD patients also display changes in their renin-angiotensin-aldosterone system (RAAS), which positively regulates ACE and angiotensin II expression, potentially aggravating SARS-CoV-2 infection [258]. Additionally, the remodeling and tissue repair caused by COPD alters ACE2 expression in epithelial cells [259]. SARS-CoV-2 has a spike protein (S) in its envelope that is activated for ACE-2 binding by serine protease TMPRSS2 (transmembrane protease, serine 2)-mediated cleavage. This protease is essential for viral infectivity and pathogenesis. The action of TMPRSS2 on the envelope protein facilitates viral entry into cells by facilitating the association with the ACE-2 receptor [260].

There are no definitive data on how COPD patient health should be managed during the current COVID-19 pandemic. Nevertheless, patients should be encouraged to continue standard therapy with inhaled corticosteroids and bronchodilators [261]. In COPD patients that develop COVID-19, the use of corticosteroids is controversial. In addition to uncertainty about their effectiveness, some studies claim that corticosteroids are contraindicated [262]. However, dexamethasone has shown a decrease in mortality in patients with COVID-19 [206], which seems to suggest their continued use for treatment of patients with pre-existing COPD who develop COVID-19.

## 8. Prevention

Quitting smoking is the best way to prevent COPD progression [182] and pathologies related to COPD, such as lung infections, lung cancer, and cardiovascular disease [20]. COPD patients report low self-esteem, low motivation to smoking cessation, and depression. Interventions that help the patient to smoking cessation, such as treatment with varenicline or bupropion (drugs that act on nicotine dependence) are fascinating [20]. Secondary prevention includes increasing physical activity in daily life, effectively preventing morbidity and mortality in COPD patients [20]. Increasing physical activity by at least 600 steps per day is associated with decreased hospitalization and COPD patient admission [263]. Secondary prevention also includes a healthy and nutritious diet, such as the Mediterranean diet, which has protective effects against respiratory diseases [264], possibly because the Mediterranean diet incorporates a balanced lipid composition with low inflammatory potential [265].

Vaccination prevents diseases in the overall population at any age. In COPD patients, vaccine in conjunction with smoking cessation, increased physical activity, and a healthy diet can improve quality of life and prevent onset of comorbidities. Influenza virus infections cause increased morbidity and mortality in COPD patients. Evidence shows decreased risk of exacerbations in patients who received influenza vaccination as a means of prevention [266]. The pneumococcal vaccine helps to keep the disease stable if administered early upon development of COPD [267].

## 9. Methodology

An initial literature research was performed by searching in the database Clinical Trials with the keyword COPD. All comparative or complementary articles as well as those with closed, retired, or unknown recruitment status were excluded. Articles selected for inclusion in this work were randomized and masked and were described in two tables, one for those intended for COPD treatment and one for those intended for COPD viral and bacterial exacerbation treatment or those aimed at improving acute exacerbations. Figure 4 shows how studies of current clinical treatments for COPD were carried out. The results of these studies are shown in the tables below.

## 10. Conclusions

COPD presents a high mortality rate because it causes organ damage and alters lung function. An imbalance between oxidants and antioxidants is a primary characteristic of COPD. Oxidative stress plays a critical role in the inflammatory response in the lungs, leading to the activation of transcription factors that amplify the inflammatory response with cell infiltration and activation and inflammatory mediators’ production. Current therapy consists of inhaled corticosteroids, bronchodilators, and both of them together. However, this is not fully effective in treating COPD or prevent exacerbations. Thus, studies aiming at the development or repurposing of new effective molecules are vital to treating COPD. Therapies to decrease oxidative stress and inflammation may improve lung function and increase patient survival. Herein, we discussed approaches focusing on prevention and treatment at a molecular level. Certain therapies, including various natural or synthetic antioxidants, can be effective because they can attenuate mucus hypersecretion, inflammation, matrix remodeling, and corticosteroid resistance. Current clinical and pre-clinical treatments under analysis include: (1) inhibitors of inflammatory mediators, phosphodiesterase, metalloprotease, neutrophil elastase, lipoxygenases, intracellular pathways (p38 MAPK, kinase inhibitor, and PI3K), EGFR, and HMG-CoA; (2) antagonist of mAChR, CRTH2, AT1R, CCR2, epithelial cells, sodium channels, CXCR2, bradykinin B1, and adenosine receptors; (3) new LAMA and LABA compounds; (4) agonists of ADRB2, lipocortin synthesis, RAR, and PPAR; and (5) stem cells therapies, immunostimulants, and gene therapies. The major challenge in COPD or exacerbation treatment is the diversity of COPD origin and time frame of intervention, too soon versus too late. Therefore, novel treatments focusing on antioxidant and anti-inflammatory activities, a new bronchodilator, a particular population cohort, targeting COPD at early or late stages, and lifestyle changes could provide new possibilities for the treatment or prevention of this noxious disease.

## Figures and Tables

**Figure 1 pharmaceuticals-14-00979-f001:**
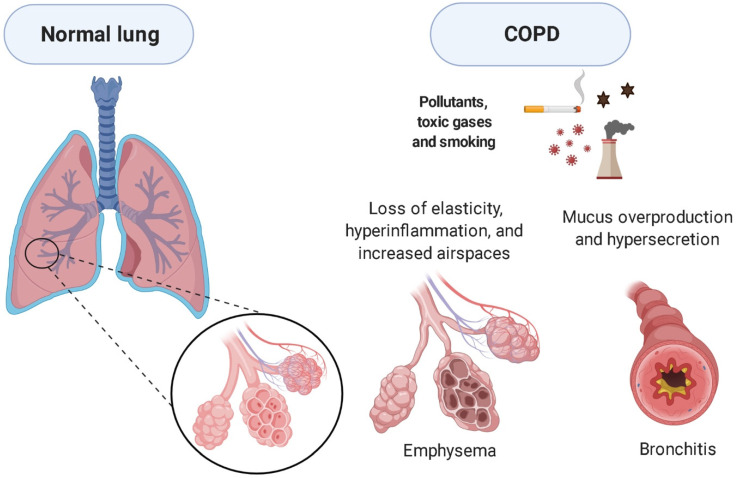
COPD phenotypes. Morphological differences exist between a normal lung and a lung with COPD. In addition, lungs with COPD can present two different characteristics: emphysema, which promotes alveolar destruction and consequent reduction in lung function, and bronchitis, which increases mucus production, narrowing airways and reducing air flow. Created with BioRender.com.

**Figure 2 pharmaceuticals-14-00979-f002:**
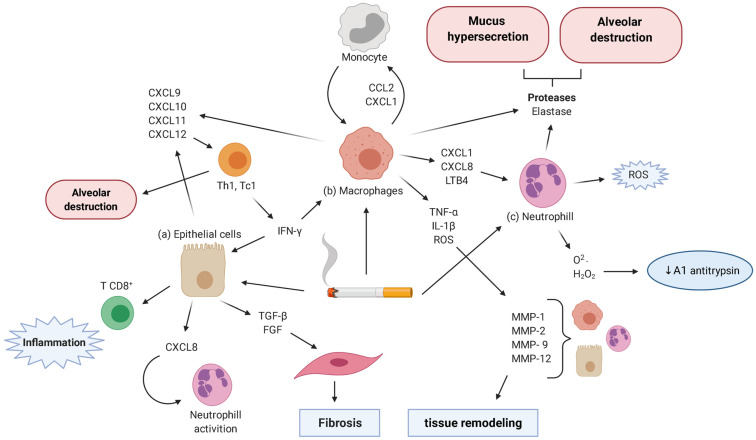
COPD pathophysiology. The toxins present in cigarette smoke lead to the recruitment of inflammatory cells and the release of inflammatory mediators. Macrophages release CXCL1, CXCL8, and LTB4, which attract neutrophils, and CCL2 and CXCL1, which attract monocytes. Neutrophils release ROS, enhancing inflammation and reductively inactivating α1 antitrypsin. They also release proteases, such as NE, leading to tissue damage. Epithelial cells and macrophages release CXCL9, CXCL10, CXCL11, and CXCL12, which attract Th1 and Tc1 lymphocytes. They also release IFN-γ, leading to alveolar destruction. Epithelial cells release CXCL8, recruiting and activating neutrophils, and TGF-β and FGF, recruiting fibroblasts that promote tissue fibrosis. Epithelial cells also attract CD8^+^ T cells believed to foster inflammation. Macrophages release TNF-α, IL-1β, and ROS, inducing MMP secretion by epithelial cells, macrophages, and neutrophils, causing tissue remodeling. CXCL1, chemokine (C-X-C motif) ligand 1; LTB4, leukotriene B4; CC, Chemokine (C-C motif); IFNγ, interferon gamma; TGF, transforming growth factor; FGF, fibroblast growth factor; TNF-α, tumor necrosis factor alpha; IL, interleukins; ROS, reactive oxygen species; MMPs, metalloproteinases. Created with BioRender.com.

**Figure 3 pharmaceuticals-14-00979-f003:**
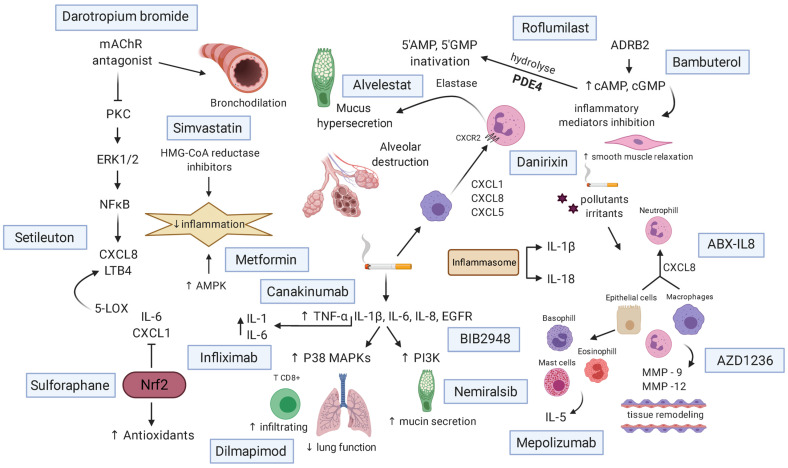
Mechanism of action of drugs for the treatment of COPD. Pollutants and CS initiate an inflammatory response by attracting inflammatory cells and releasing inflammatory mediators. mAChR antagonists act as bronchodilators, promoting the release of CXCL8 and LTB4. AMPK and HMG-CoA reductase stimulants decrease inflammation. Inhibitors of inflammatory mediators, such as CXCL1, CXCL8, and CXCL5, act by decreasing chemoattraction of neutrophils and macrophages to the lung (underlying inflammation). Nrf2 stimulants increase the transcription of antioxidant genes, and block the release of pro-inflammatory mediators. Interleukin, EGFR, and TNF-α inhibitors antagonize the activation of MAPKs and PI3K, and attenuate the release of pro-inflammatory mediators, such as IL-6 and IL-1. ADRB2 agonists inhibit the release of inflammatory mediators, cause smooth muscle relaxation, and increase cAMP and cGMP. PDE4 inhibitors prevent cAMP degradation, increasing intracellular cAMP levels, leading to smooth muscle relaxation, and enhancing the bronchodilator effects of β-agonists. CS, cigarette smoke; mAChR, Muscarinic ACh receptors AMP, adenosine monophosphate, AMPK, AMP-activated protein kinase; ADRB2, beta-2-adrenergic receptor; CXCL, chemokine; IL, interleukin; ERK, extracellular signal-regulated kinases; GMP, guanosine monophosphate; HMG-CoA, 3-hydroxymethylglutaryl CoA reductase; LOX, lipoxygenase; LTB4, leukotriene B4; MMPs, matrix metalloproteinases; ADRB2, adrenoceptor Beta 2; NFkB, factor nuclear kappa B; Nrf2, nuclear factor erythroid 2-related factor 2; PDE4, phosphodiesterase 4, PI3K, phosphoinositide 3-kinases; PKC, protein kinase C; TNF-α, tumor necrosis factor alpha. Created with BioRender.com.

**Figure 4 pharmaceuticals-14-00979-f004:**
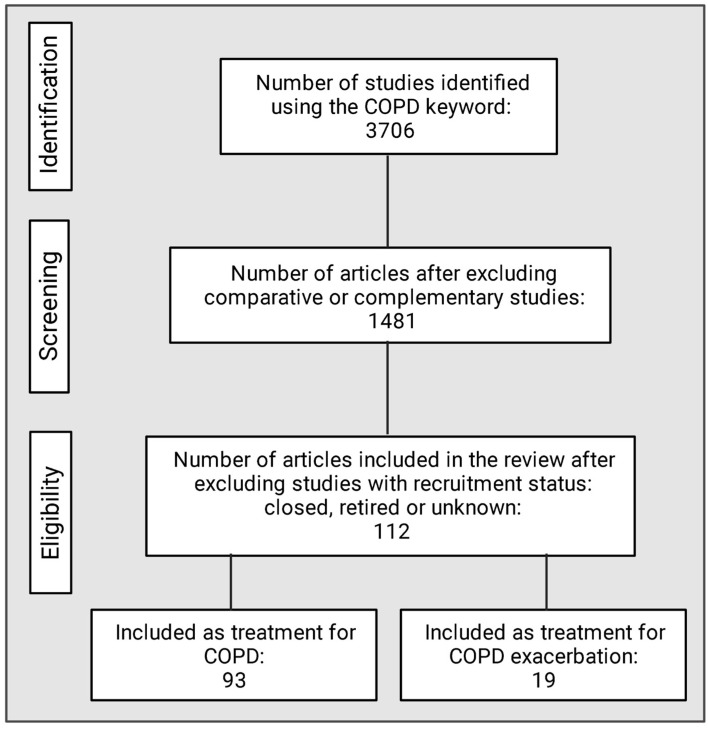
Research methodologies used to identify COPD treatments. Created with BioRender.com.

**Table 1 pharmaceuticals-14-00979-t001:** Updating drug research for COPD treatment.

Drug Other Names	Use	Target/Action	Identification	Phase	Status	Approved COPD Treatment	Approved Treatment for Another Disease
Canakinumab ACZ885	Antibody	IL-1β inhibitor	NCT00581945	1/2	Completed		Adult-onset Still’s disease, Gouty arthritis, and others
ABX-IL8	Antibody	IL-8 inhibitor	NCT00035828	2	Completed		
Infliximab Remicade, TA-650	Antibody	TNF-α inhibitor	NCT00056264	3	Completed		Ankylosing spondylitis, Crohn’s disease, and others
Mepolizumab	Antibody	IL-5 inhibitor	NCT04075331	2/3	Recruiting		Asthma
SB-240563, Bosatria, Nucala			NCT01463644	3	Completed		
Ensifentrine	Uninformed	PDE3/PDE4 inhibitor	NCT03443414	2	Completed		
RPL554			NCT04091360	2	Completed		
Roflumilast Daliresp	Oral	PDE4 inhibitor	NCT01509677	3	Completed	Yes	
AZD2115	Inhaled	MABA	NCT01498081	2	Completed		
			NCT02109406	2	Completed		
MEDI8968 AMG-108		IL-1 antagonist	NCT01448850	2	Completed		
Benralizumab	Antibody	IL-5 inhibitor	NCT01227278	2	Completed		Asthma
			NCT04053634	3	Recruiting		
AZD1236		MMP-9/12 inhibitor	NCT00758706	2	Completed		
Glycopyrrolate SUN-101, Glycopyrrolate bromide, glycopyrronium bromide, NVA-237, AD-237 Seebri Breezhaler, CHF-5259	Inhaled	M3 receptor antagonists	NCT00545311	1	Completed	Yes	
			NCT00242333	2	Completed		
			NCT00856193	2	Completed		
			NCT02680197	2	Completed		
			NCT02189577	2	Completed		
			NCT02347761	3	Completed		
			NCT01566604	3	Completed		
			NCT01154127	3	Completed		
			NCT01715298	3	Completed		
			NCT01005901	3	Completed		
			NCT02371629	4	Completed		
Glycopyrrolate-formoterol Bevespi Aerosphere	Inhaled	MABA	NCT01854645	3	Completed	Yes	
			NCT01854658	3	Completed		
			NCT01970878	3	Completed		
Glycopyrrolate-indacaterol Utibron Neohaler	Inhaled	MABA	NCT01727141	3	Completed	Yes	
			NCT01712516	3	Completed		
			NCT01682863	3	Completed		
Simvastatin	Oral	HMG-CoA reductase inhibitors Nitric oxide synthase type II Inhibitors IL-17	NCT01944176	3	Completed		Diabetic cardiomiopathy, HCL ^1^, hyperlipidemia, and others
			NCT02070133	3	Completed		
Rhodiola Crenulata	Oral	Anti-inflammation and anti-oxidation	NCT02242461	2	Completed		
Revefenacin GSK1160724, TD-4208	Inhaled	mAChR antagonist	NCT00555022	1	Completed	Yes	
			NCT02040792	2	Completed		
			NCT02109172	2	Completed		
			NCT02512510	3	Completed		
Sulforaphane SFX-01, Broccoli-sprout-extract	Oral	Nrf2 stimulator	NCT01335971	2	Completed		
QVA149 Indacaterol maleate/ glycopyrronium bromide, NVA-237/QAB-149	Inhaled	MABA	NCT01996319	3	Completed	Yes	
			NCT01120717	3	Completed		
Indacanterol	Inhaled	LABA	NCT00636961	2	Completed	Yes	
			NCT00792805	3	Completed		
			NCT01543828	4	Completed		
CHF6523	Uninformed	PI3K inhibitor	NCT04032535	1	Recruiting		
AZD8683	Inhaled	M3 receptor antagonist	NCT01205269	2	Completed		
Tiotropium bromide Ba 679 BR, Spiriva, PUR-0200	Inhaled	M1 and M3 receptor antagonist	NCT01921712	1	Completed	Yes	Asthma
			NCT02671825	1	Completed		
			NCT02172352	2	Completed		
			NCT02175342	2	Completed		
			NCT00292448	2	Completed		
			NCT02172391	3	Completed		
			NCT00144339	3	Completed		
			NCT00274573	3	Completed		
			NCT00274547	3	Completed		
			NCT02172378	3	Completed		
			NCT00274053	3	Completed		
			NCT00168831	3	Completed		
			NCT04061161	4	Recruiting		
			NCT00523991	4	Completed		
			NCT01072396	4	Completed		
			NCT00274079	4	Completed		
Bambuterol Bambec, KWD 2183	Oral	ADRB2 agonist	NCT01796730	4	Completed	Yes	Asthma and bronchitis
AZD3199	Inhaled	ADRB2 agonist	NCT00929708	2	Completed		
PT003 Formoterol/ glycopyrrolate, PT005/PT001, GFF MDI, Bevespi	Inhaled	MABA	NCT04087590	2	Recruiting	Yes	
			NCT02347085	3	Completed		
			NCT02643082	3	Completed		
Astegolimab MSTT1041A, AMG 282, Anti-ST2, RO 7187807	Antibody	IL-33 inhibitor	NCT03615040	2	Completed		
AZD1981	Oral	CRTH2 antagonist	NCT00690482	2	Completed		
Abetiterol LAS100977, AZD-0548	Inhaled	LABA	NCT01425814	2	Completed		
			NCT01425801	2	Completed		
Formoterol fumarate CHF 1531	Inhaled	LABA	NCT00215436	3	Completed	Yes	Asthma
CHF 6001 Tanimilast	Inhaled	PDE4 inhibitor	NCT01703052	1	Completed		
			NCT02386761	1	Completed		
			NCT01730404	2	Completed		
			NCT03004417	2	Completed		
DNK333	Uninformed	NK1/NK2 antagonist	NCT01287325	½	Completed		
Aclidinium Bromide LAS 34273, KRP-AB1102, Bretaris Genuair, Eklira Genuair, Tudorza	Inhaled	M3 receptor inhibitor	NCT03276052	1	Not yet recruiting	Yes	
			NCT01471171	3	Completed		
			NCT00970268	3	Completed		
			NCT00891462	3	Completed		
			NCT00358436	3	Completed		
			NCT00500318	3	Completed		
			NCT01966107	4	Completed		
Quercetin	Oral	Inflammation and oxidative stress	NCT01708278	1	Completed		
			NCT03989271	1/2	Recruiting		
Arformoterol tartrate Brovana	Inhaled	LABA	NCT00691405	2	Completed	Yes	
			NCT00250679	3	Completed		
			NCT00909779	3	Completed		
BIO-11006	Inhaled	MARCKS inhibitor	NCT00648245	2	Completed		
Bimosiamose	Inhaled	Pan-selectin antagonist	NCT01108913	2	Completed		
QBW251	Oral	CFTR stimulant	NCT02449018	2	Completed		
			NCT04268823	2	Recruiting		
Bufei Jianpi granule	Oral	Elaying pulmonar function decline	NCT03976700	3	Not yet recruiting		
Alvelestat MPH966, AZD9668	Oral	Neutrophil elastase inhibitor	NCT01035411	1	Completed		
			NCT03679598	2	Recruiting		
			NCT00703391	2	Completed		
			NCT01054170	2	Completed		
Tetomilast OPC-6535	Oral	PDE4 inhibitor	NCT00917150	2	Completed		
Ipratropium Bromide	Inhaled	LAMA	NCT02236182	2	Completed	Yes	
			NCT00202176	4	Completed		
Setileuton MK-0633		5-LOX inhibitor	NCT00418613	2	Completed		
Cyclosporine	Oral	Calcineurin inhibitor	NCT00974142	1/2	Completed		
PT010 Budesonide/ formoterol/glycopyrrolate, BGF-MDI, Budesonide/PT 003	Inhaled	ICS/LAMA/ LABA	NCT03906045	1	Completed	Yes	
Lovastatin	Oral	HMG-CoA reductase inhibitor	NCT00700921	2	Completed		HCL ^1^ and hyperlipidemia
Reldesemtiv CK-107, CK-2127107	Oral	Troponin stimulant	NCT02662582	2	Completed		
Symbicort Budesonide-formoterol	Inhaled	ICS/LABA	NCT00206154	3	Completed	Yes	Asthma, Crohn’s disease, and ulcerative colitis
			NCT00206167	3	Completed		
Rosuvastatin	Oral	HMG-CoA reductase inhibitor	NCT00929734	2	Completed		Atherosclerosis, cardiovascular disorders, HCL^1^, and others
Dilmapimod GSK 681323, SB681323	Uninformed	p38 MAPK inhibitor	NCT00144859	2	Completed		
Losartan	Oral	AT1 receptor antagonist	NCT00720226	4	Completed		Diabetic nephropathies, heart failure, and hypertension
			NCT02696564	4	Active, not recruiting		
Levalbuterol Xopenex HFA	Inhaled	SAMA	NCT00665600	3	Completed	Yes	Asthma
Albuterol Salbutamol	Spray aerosol, injectable or inhaled	SABA	NCT00440245	4	Completed	Yes	Asthma
Albuterol-ipratrópio Combivent Respimat	Inhaled	MABA	NCT00400153	3	Completed	Yes	
AZD2423	Oral	CCR2 antagonist	NCT01153321	2	Completed		
PH-797804	Oral	p38 MAPK inhibitor	NCT00559910	2	Completed		
			NCT01321463	2	Completed		
CHF6366	Inhaled	MABA	NCT03378648	1/2	Completed		
Indacaterol Arcapta	Inhaled	LABA	NCT00624286	3	Completed	Yes	
MEDI2338 CERC 007	Intravenous	IL-18 inhibitor	NCT01322594	1	Completed		
AZD5069	Oral	CXCR2 antagonist	NCT01233232	2	Completed		
UMC119-06	Intravenous	Cell replacements	NCT04206007	1	Recruiting		
ION-827359	Inhaled	Epithelial sodium channel antagonist	NCT04441788	2	Recruiting		
Erdosteine	Oral	Glycoprotein inhibitor	NCT00338507	2	Completed	Yes	Bronchitis
RV1162 PUR 1800	Inhaled	Narrow-spectrum kinase inhibitor	NCT01970618	1	Completed		
JNJ 49095397 RV568	Inhaled	PTS inhibitors	NCT01867762	2	Completed		
Selenium	Oral	GPx-1 levels	NCT00186706	4	Completed		
Epeleuton DS102, 15-HEPE, AF-102	Oral	5-LOX inhibitor	NCT03414541	2	Completed		
AZD8871	Inhaled	MABA	NCT02814656	1	Completed		
			NCT02971293	2	Completed		
CHF 5993 Beclometasone/formoterol/glycopyrrolate, BDP/FF/GB	Inhaled	ICS/LABA/ LAMA	NCT02743013	1	Completed	Yes	Asthma
Vilanterol GW642444	Inhaled	LABA	NCT00372112	2	Completed		
			NCT00606684	2	Completed		
GSK256066	Inhaled	Type 4 cyclic nucleotide PDE inhibitors	NCT00549679	2	Completed		
PF00610355	Inhaled	ADRB2 agonist	NCT00808288	2	Completed		
Umeclidinium GSK573719, Incruse Ellipta	Inhaled	LAMA	NCT01110018	1	Completed	Yes	
			NCT00950807	2	Completed		
			NCT00732472	2	Completed		
			NCT01030965	2	Completed		
			NCT01387230	3	Completed		
			NCT02184611	3	Completed		
Umeclidinium-vilanterol Anoro Elipta	Inhaled	MABA	NCT01899742	3	Completed	Yes	
Darotropium bromide GSK233705	Inhaled	mAChR antagonist	NCT00676052	2	Completed		
			NCT00376714	2	Completed		
			NCT00453479	2	Completed		
Tiotropium-Olodaterol Stiolto Respimat	Inhaled	MABA	NCT01431274	3	Completed	Yes	
			NCT01431287	3	Completed		
AZD8871	Inhaled	MABA	NCT03159442	1	Completed		
Oglemilast GRC 3886, GRC-3836	Oral	PD4 inhibitor	NCT00671073	2	Completed		
Danirixin GSK-1325756	Oral	CXCR2 antagonist	NCT01209052	1	Completed		
			NCT03034967	2	Completed		
			NCT02130193	2	Completed		
Fluticasone Propionate/ salmeterol Advair HFA	Inhaled	ICS/LABA Arachidonic acid inhibitors Lipocortin synthesis agonists	NCT00633217	4	Completed	Yes	Asthma
Batefenterol GSK961081	Inhaled	MABA	NCT00887406	1	Completed		
			NCT02663089	1	Completed		
			NCT00478738	2	Completed		
			NCT02570165	2	Completed		
Remestemcel-L RYONCIL	Intravenous	Stem cell therapies	NCT00683722	2	Completed		Graft-versus-host disease
Budesonida Pulmicort	Inhaled	ICS	NCT00232674	4	Completed	Yes	Asthma
N-acetylcysteine	Oral	Antioxidant	NCT02579772	4	Completed	Yes	Bronchiectasis, cystic fibrosis, dry eyes, and poisoning
MK-0873	Oral	PDE4 inhibitor	NCT00132730	2	Terminated		
BI 1026706	Oral	Bradykinin B1 receptor antagonist	NCT02642614	1	Completed		
BIBW 2948	Inhaled	EGFR inhibitor	NCT00423137	2	Completed		
Cilomilast SB 207499, AL-38583	Oral	PDE4 inhibitor	NCT00103922	3	Completed		
Olodaterol BI 1744 CL, Striverdi Respimat	Inhaled	LABA	NCT00824382	2	Completed	Yes	
			NCT00452400	2	Completed		
			NCT01809262	2	Completed		
			NCT00793624	3	Completed		
PH-797804	Oral	p38 inhibitor	NCT01543919	2	Completed		
CCI15106	Inhaled	Undefined mechanism	NCT03235726	1	Completed		
Losmapimod GW856553X, FTX-1821	Oral	p38α/β MAPK inhibitor	NCT01218126	2	Completed		
BI 113608	Oral	Undefined mechanism	NCT01958008	1	Completed		
TRN-157	Inhaled	M3 receptor antagonists	NCT02133339	1	Completed		
PF03635659	Inhaled	Undefined mechanism	NCT00864786	1	Completed		
CNTO 6785	Intravenous	IL17A protein inhibitor	NCT01966549	2	Completed		
Sildenafil Viagra	Oral	PDE5 inhibitor	NCT00104637	2	Completed		Erectile dysfunction and pulmonary arterial hypertension
AZD7594 AZ13189620	Inhaled	Glucocorticoid receptor modulators	NCT02645253	1	Completed		
Tofimilast CP 325366	Inhaled	PDE4 inhibitor	NCT00219622	2	Completed		
Fluticasone-furoate/vilanterol	Inhaled	LABA/ICS	NCT01691885	3	Completed	Yes	Asthma
Retinoic Acid		RAR agonists	NCT00000621	2	Completed		Acne, acute promyelocytic leukaemia, photodamage, and warts
Lebrikizumab	Subcutaneous	IL-13 inhibitor	NCT02546700	2	Completed		
Fluticasone- umeclidinium Trelegy Ellipta	Inhaled	MABA	NCT02345161	3	Completed	Yes	Asthma
			NCT02729051	3	Completed		

^1^ HCL, hypercholesterolemia.

**Table 2 pharmaceuticals-14-00979-t002:** Update of drug investigation for the treatment of COPD exacerbation caused by viral and bacterial infection and directed to the treatment of acute exacerbations.

Drug Other Names	Use	Target/Action	Identification	Phase	Status	Approved COPD Treatment	Approved Treatment for Another Disease
Ciprofloxacin	Oral	DNA gyrase Inhibitor DNA topoisomerase inhibitor	NCT02300220	3	Completed		Acute sinusitis, gonorrhoea, Intestinal infections, respiratory tract infections, and others
Tezepelumab MEDI-9929	Subcutaneous	Hymic stromal lymphopoietin inhibitor	NCT04039113	2	Recruiting		
Ismigen Antibacterial vaccine sublingual, Provax, Pulmigen, Respibron, Bactovax, Bromunyl.	Sublingual	Immunostimulants	NCT02417649	4	Completed	Yes	Respiratory tract infections
Doxycycline	Oral	Protein 30S ribosomal subunit inhibitors	NCT02305940	3	Completed		
Levofloxacin MP-376, Quinsair, Aeroquin	Inhaled	DNA gyrase inhibitor DNA topoisomerase type IV and type II inhibitor	NCT00739648	2	Completed		Bacterial infections, pneumonia, Sinusitis, tuberculosis, and others
Roflumilast	Oral	PDE4 inhibitor	NCT00076089 NCT00430729	3 3	Completed	Yes	
Benralizumab	Subcutaneous	Anti-IL5Rα antibody	NCT04098718	2	Not yet recruiting		Asthma
			NCT04053634	3	Recruiting		
			NCT02155660	3	Completed		
Aclidinium Bromide LAS 34273, KRP-AB1102, Bretaris Genuair, Eklira Genuair, Tudorza	Inhaled	M3 receptor inhibitor	NCT01966107	4	Completed	Yes	
Metformin	Oral	AMPK stimulants Gluconeogenesis inhibitors	NCT01247870	4	Completed		Type 2 diabetes mellitus
Enoximone	Intravenous	PDE3 inhibitor	NCT04420455	4	Not yet recruiting		Heart failure
QBW251	Oral	CFTR stimulant	NCT04268823	2	Recruiting		
Losmapimod GW856553X, FTX-1821	Oral	p38α/β MAPK inhibitor	NCT02299375	2	Completed		
Nemiralisib GSK2269557	Inhaled	PI3Kδ inhibitor	NCT02294734	2	Completed		
Acumapimod BCT197	Oral	P38 inhibitor	NCT02700919	2	Completed		
Mepolizumab SB-240563, Bosatria, Nucala	Subcutaneous	IL-5 inhibitor	NCT04133909	3	Recruiting		Asthma
Azithromycin	Oral	Protein 50S ribosomal subunit inhibitors	NCT04319705		Recruiting		Acute exacerbations of chronic bronchitis, acute sinusitis, pneumonia, pharyngitis, and respiratory tract infections
Arbidol	Oral	DNA and RNA synthesis inhibiting	NCT03851991		Recruiting		
Anti-ST2 MSTT1041A, AMG 282	Subcutaneous	IL33 inhibitors	NCT03615040		Not yet recruiting		

**Table 3 pharmaceuticals-14-00979-t003:** Pre-clinical research methods in vivo used in new drug discovery and development.

Study	Treatment	Model Species	Experimental Intervention	Results
Horio et al., 2017	Galectina (Gal) −9 administered subcutaneously once daily from 1 day before PPE instillation to day 5	Female C57BL/6 mice (8–10 weeks old)	Lungs were intratracheally instilled with two units of PPE diluted in 50 μL of saline via 24-gauge catheter on day 0	Infiltration of neutrophils was inhibited and MMP levels decreased
Melo et al, 2018	Atorvastatin, 1, 5, and 20 mg, treated from day 33 until day 64 via inhalation for 10 min once a day	Male C57BL/6 mice (8 weeks old/18–22 g)	Administered intranasally 4 × 0.6 U of porcine pancreatic elastase (PPE) every other day (days 1, 3, 5 and 7).	Induced lung tissue repair in mice with emphysema
Pinho-Ribeiro et al., 2017	Atorvastatin and simvastatin administered via inhalation for 15 min (1 mg/mL, once/day)	Male C57BL/6 mice (8–10 weeks old)	Mice exposed to 12 cigarettes a day for 60 days, then treated for another 60 days	Improved lung repair after cigarette smoke-induced emphysema, accompanied by a reduction in oxidative stress markers.
Sun et al., 2017	Simvastatin administered intra-gastrically at a dose of 5 mg/kg/day followed by CS	Male Sprague Dawley (SD) rats (6 weeks old/110–20 g)	Animals were passively exposed (whole body) to smoke from 20 cigarettes in a box for 1 hour, twice a day, 5 days a week, for 16 weeks	Partial blockage of airway inflammation, and MMP production
Susuki et al., 2009	Curcumin (100 mg/kg) administrated daily by oral gavage throughout a 21-day period	Male C57BL/6J mice (9 weeks old)	Administered intratracheal porcine pancreatic elastase (PPE), or exposed to CS (60 min/day for 10 consecutive days, or 5 days/week for 12 weeks)	Inhibited PPE-induced increase in neutrophils, inhibited increase in neutrophils and macrophages in BAL, and attenuated increase in air space induced by CS
Kennedy-Feitosa et al., 2018	Inhalation of 1 mg/mL or 10 mg/mL eucalyptol for 15 min per day	Male C57BL/6 mice (8 weeks old/18–25 g)	Mice exposed to 12 cigarettes a day for 60 days, then treated for another 60 days without exposure to smoke	Lung repair, reduced inflammatory cytokines and NE levels, and increased elastin and TIMP-1 levels.
Boo et al., 2020	LJ-2698 (50 μg/kg) administrated by oral gavage six times per week for 5 weeks.	FVB mice (8 weeks old)	One week after drug treatment, 0.25 units of PPE was intratracheally instilled into the lungs of the mice	Induction of anti-inflammatory cytokine production and recruitment of M2 macrophages

## Data Availability

Not applicable.

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
