# Peer review of "Mechanisms, Pathophysiology and Currently Proposed Treatments of Chronic Obstructive Pulmonary Disease"

_pharmaceuticals, 2021, doi:10.3390/ph14100979_

Round 1

Reviewer 1 Report

GENERAL COMMENTS

This is a review article in which the authors have provided an overview on potential new therapeutic strategies in patients with COPD. An attempt to review therapies on the basis of COPD pathophysiology has been made. However, at some point the authors have failed to do so. Despite the potential interest of the study question I have several important concerns that are summarized below.

SPECIFIC MAJOR COMMENTS

1) Check affiliations, since # 3 is missing and first author cannot bear affiliation # 6. Organize the number of affiliations.

2) Title is misleading, since information on lung function or mechanics, which are major contributors to dyspnea and/or major clinical symptoms, has not been provided at all in the review.

3) The review has focused exclusively on the underlying biology taking place in the airways and lungs of patients with COPD. However, no information has been given on lung function deterioration or lung mechanics modifications, which substantially contribute to most clinical symptoms including exercise intolerance in these patients. Moreover, most of currently available drugs are based on the pathophysiology taking place in the airways and lungs in COPD. No mention to any of these paramount aspects has been made. In my view, this is a major concern of this review.

4) In my opinion, the review should have addressed aspects at different levels: respiratory system, organs, and cells. The first two levels have not been addressed at all in the review. Section # 3 is very poor in this respect.

5) Section # 5 has also been very poorly described. How genetic and epigenetic regulation are involved in COPD patients? To what extent these mechanisms can help design therapeutic strategies? For instance, line # 311 “undergo mutations”: do they induce pathology/functional alterations? In which organ/cell types?

6) Lines # 333 & 334: what do they mean precisely?

7) Lines # 373 and below. The authors seem to have disregarded the contribution of bacteria and virus to COPD exacerbations as well as other major causes (e.g. pulmonary thromboembolism).

8) Tables 1-3 are totally useless and should be placed, if any use can be found for them, in a supplement/appendix. They are extremely long, contain irrelevant and misleading information. Moreover, they represent a mere list of names of drugs with no explanations provided on the roles or precise indications of each drug in patients with COPD. Anyone could find this information in different web sites. This is not information to be included in a comprehensive review article.

9) Information in Table 3 refers to animal studies? This must be indicated.

10) Lines 510 & 511, this could be applied to any person, not only to COPD.

11) Conclusions are irrelevant and not based on a proper review of the existing literature as to availability of different strategies in COPD patients.

12) The influence of COPD in organs and systems beyond the lungs and airways and/or comorbidities that are frequent in these patients, have not been addressed at all in the review.

Author Response

REVIEWER GENERAL COMMENTS

This is a review article in which the authors have provided an overview on potential new therapeutic strategies in patients with COPD. An attempt to review therapies on the basis of COPD pathophysiology has been made. However, at some point the authors have failed to do so. Despite the potential interest of the study question I have several important concerns that are summarized below.

SPECIFIC MAJOR COMMENTS

1) Check affiliations, since # 3 is missing and first author cannot bear affiliation # 6. Organize the number of affiliations.

Response: We organized the number of affiliations.

2) Title is misleading, since information on lung function or mechanics, which are major contributors to dyspnea and/or major clinical symptoms, has not been provided at all in the review.

Response: We thank the Reviewer for this suggestion. We realize that pathophysiology is a very broad term to be used in COPD discussion. We changed the title and inserted three new paragraphs into section #3. We described the normal dynamics of alveolar septa in terms of extracellular matrix, how they are destroyed, and the consequent inflammatory process leading to changes in mechanical properties the lungs. We further extended the pathophysiology discussion, focusing not only on the lungs but also on the heart (right ventricle) and diaphragm dysfunction. We associated their impairment with increased exacerbation and hospitalization levels, which may contribute to a high mortality rate.

3) The review has focused exclusively on the underlying biology taking place in the airways and lungs of patients with COPD. However, no information has been given on lung function deterioration or lung mechanics modifications, which substantially contribute to most clinical symptoms including exercise intolerance in these patients. Moreover, most of currently available drugs are based on the pathophysiology taking place in the airways and lungs in COPD. No mention to any of these paramount aspects has been made. In my view, this is a major concern of this review.

Response: We thank the Reviewer for this important question. We inserted three new paragraphs correlating lung function decline with clinical worsening. We further discussed the association of COPD with heart function (pulmonary arterial hypertension due to respiratory conditions) and with the diaphragm. Most pharmacologic agents such as corticosteroid and LABA agents have effects on exacerbation prevention since exacerbation is directly related to increased mortality.

4) In my opinion, the review should have addressed aspects at different levels: respiratory system, organs, and cells. The first two levels have not been addressed at all in the review. Section # 3 is very poor in this respect.

Response: We thank the Reviewer for this important matter. We included the clinical impact of lung function decline as also the association of COPD with PAH. COPD patients who present PAH have decreased survival rate compared to COPD patients at similar severity but without PAH. In addition, we discussed a likely explanation for that association.

5) Section # 5 has also been very poorly described. How genetic and epigenetic regulation are involved in COPD patients? To what extent these mechanisms can help design therapeutic strategies? For instance, line # 311 “undergo mutations”: do they induce pathology/functional alterations? In which organ/cell types?

Response: Absolutely. The section had been poorly described. We just had cited a few examples improperly. We rewrote that section to better describe the previously quoted articles and increased the number of references to better discuss how genetic and epigenetic regulation are involved in COPD patients. We related those mechanisms to the design of therapeutic strategies. We discussed the epigenetic regulatory role in COPD phenotype changes, pathogenesis, prognostic as biomarkers, and therapy. We also associated some mutations to pathology/functional alterations in specific organ/cell types, especially in the lung, as a primary concern in COPD.

6) Lines # 333 & 334: what do they mean precisely?

Response: The drugs currently used are inhaled bronchodilators and corticosteroids. However, COPD is a complex disease affecting many cells. Therefore, it is interesting to have new drugs that target the relief of symptoms and modulate the pathophysiology. After robust research, we concluded that there is a search for new drugs in pre-clinical trials.

7) Lines # 373 and below. The authors seem to have disregarded the contribution of bacteria and virus to COPD exacerbations as well as other major causes (e.g., pulmonary thromboembolism).

Response: The focus was on viral and bacterial exacerbations, and we agree it seemed we had disregarded bacteria and virus to COPD exacerbations and other major causes. We modified the text to clarify that matter.

8) Tables 1-3 are totally useless and should be placed, if any use can be found for them, in a supplement/appendix. They are extremely long, contain irrelevant and misleading information. Moreover, they represent a mere list of names of drugs with no explanations provided on the roles or precise indications of each drug in patients with COPD. Anyone could find this information in different web sites. This is not information to be included in a comprehensive review article.

Response: We added more informative columns in the tables, information in the text about drugs included in the table, and their form of action. We added more helpful information to guide the reader better. We have included the form of administration, whether or not the drug is approved for treatment in other diseases, and the current status in the tables.

9) Information in Table 3 refers to animal studies? This must be indicated.

Response: Table 3 reports on different animal methodologies used to research new drugs for COPD, and as requested, we indicate that it refers to the use of animals. 

10) Lines 510 & 511, this could be applied to any person, not only to COPD.

Response: We agree this could be applied to everybody and added a sentence. Vaccination is a form of protection against several serious diseases and their complications. In COPD, individuals should be encouraged to vaccinate to reduce the risk of comorbidities associated with the disease. In this review, we chose to provide this information on prevention to stimulate vaccination in patients.

11) Conclusions are irrelevant and not based on a proper review of the existing literature as to availability of different strategies in COPD patients.

Response: We agree the conclusion was vague, not based on the proper review of the existing literature. We rewrote the conclusion.

12) The influence of COPD in organs and systems beyond the lungs and airways and/or comorbidities that are frequent in these patients, have not been addressed at all in the review.

Response: We inserted three new paragraphs into section #3, including information about COPD causing diaphragm and heart dysfunction, and comorbidities into section #1 and #6.

Reviewer 2 Report

The authors present a very good review on the topic on COPD pathophysiology and how it can be targeted by the current and future therapies. This is a really well documented and structured revier, that migth benefit from a few minor improvements and corrections:

  1. Introduce more COVID-19 disease context in line 79, as an introduction of this new infectious disease, and why you are reporting its relation to COPD.
  2. In the section of epigenetic modulations in COPD, please at least mention that also there occurs a dysregulation in the expression of noncoding RNAs such as microRNAs that is caused by exposure to cigarette smoke.
  3. In relation to this previous point, it would also be of interest to mention briefly that epigenetics deregulated by cigarette smoke are able to modulate mucus related genes, and that targeted epigenetic editing might also be a possible therapeutic tool in this matter (reviewed previously in doi: 10.1165/rcmb.2017-0072TR)
  4. It would also be interesting to introduce some more specific information on current and possible treatments that may control mucus secretion and clearance, and emphasize how some of the currently mentioned treatments have also been shown to downregulate mucin gene expression.

Formatting revisions.

  1. In line 230 there is a reference that is not related to any number (KIRKHAM, 2007), please revise.
  2. In line 281 there is a reference that is not related to any number (BARNES et al., 2016), please revise.
  3. In lines 282, 423 and 435, the paragraph starts too much aligned to the left, please revise.

Author Response

The authors present a very good review on the topic on COPD pathophysiology and how it can be targeted by the current and future therapies. This is a really well documented and structured revier, that migth benefit from a few minor improvements and corrections:

  1. Introduce more COVID-19 disease context in line 79, as an introduction of this new infectious disease, and why you are reporting its relation to COPD.

Response: Thanks for the nice words. We appended more information to contextualize the relation between COVID-19 and COPD better. We had not added enough information.

  1. In the section on epigenetic modulations in COPD, please at least mention that there is also a dysregulation in the expression of noncoding RNAs such as microRNAs caused by exposure to cigarette smoke.

Response: We definitely should have mentioned and discussed microRNAs. In the revised version of the manuscript, we point out the dysregulation in the expression of microRNAs induced by cigarette smoke exposure in the section of epigenetic modulations in COPD. We discussed the epigenetic regulatory role in COPD phenotype changes, pathogenesis, prognostic as biomarkers, and therapy.

  1. In relation to this previous point, it would also be of interest to mention briefly that epigenetics deregulated by cigarette smoke are able to modulate mucus related genes, and that targeted epigenetic editing might also be a possible therapeutic tool in this matter (reviewed previously in doi: 10.1165/rcmb.2017-0072TR)

Response: Using the suggested review written by Saco et al. 2018, we highlighted the mucus-related genes are the focus of deregulated epigenetics by cigarette and reinforced the role of targeted epigenetic editing as a possible therapeutic tool. We also added other references to this section.

  1. It would also be interesting to introduce some more specific information on current and possible treatments that may control mucus secretion and clearance, and emphasize how some of the currently mentioned treatments have also been shown to downregulate mucin gene expression.

Response: We added information about drugs that are mentioned in the table and their form of action, including those that target the mucin coding gene.

Formatting revisions.

  1. In line 230 there is a reference that is not related to any number (KIRKHAM, 2007), please revise.
  2. In line 281 there is a reference that is not related to any number (BARNES et al., 2016), please revise.
  3. In lines 282, 423 and 435, the paragraph starts too much aligned to the left, please revise.

Response: Thanks for the careful revision. The texts were revised and those needed changes were made.

Reviewer 3 Report

This is a very nice overview of Drugs used in the treatment ( and in development) for COPD

Overall comment is that there is some generalization of mechanisms associated with COPD and it is imperative to note that the taxonomy of COPD is evolving focusing on differentiating early or pre-COPD from different types of COPD that may be associated with some of the mechanisms described. It is plausible that most anti-inflammatory trials failed to demonstrate an effect because they did not target the right subset of patients or pathways that have a role in early vs late phase of the disease (too little, too late).

Specific

  • Line 220-222 seems to describe an endotype (mechanistic subset) not a phenotype. See Agustí A, Celli B, Faner R. What does endotyping mean for treatment in chronic obstructive pulmonary disease? Lancet. 2017 Sep 2;390(10098):980-987. doi: 10.1016/S0140-6736(17)32136-0. PMID: 28872030.

  • Table 1
    • Fevipiprant, BIO-11006, losmapimod, nemiralisib, lebrilizumab are all terminated  - The authors should carefully review the therapies do determine their current status ( and then note in the table/paper when this was done- as this changes quickly)
    •  
    • Remove Relovair as trademark names are different globally
    • Some of the therapies listed are actually approved in the treatment of other diseases.  Perhaps  an additional column or some type of notation could indicate this.

  • Table 2 studies most completed and reported

  • May be useful to add inhaled, oral, antibody to Tables 1&2

  • Line 497 Smoking cessation or tobacco treatment (vs quitting smoking)

Author Response

This is a very nice overview of Drugs used in the treatment ( and in development) for COPD

Overall comment is that there is some generalization of mechanisms associated with COPD and it is imperative to note that the taxonomy of COPD is evolving focusing on differentiating early or pre-COPD from different types of COPD that may be associated with some of the mechanisms described. It is plausible that most anti-inflammatory trials failed to demonstrate an effect because they did not target the right subset of patients or pathways that have a role in early vs late phase of the disease (too little, too late).

Response: Thanks for the compliment and for the cutting-edge view.

Specific

  • Line 220-222 seems to describe an endotype (mechanistic subset) not a phenotype. See Agustí A, Celli B, Faner R. What does endotyping mean for treatment in chronic obstructive pulmonary disease? 2017 Sep 2;390(10098):980-987. doi: 10.1016/S0140-6736(17)32136-0. PMID: 28872030.

Response: We intended to approach polarization of macrophages due to the cells and inflammatory mediators found at the site. We understood it is not called phenotype and changed the sentences.

  • Table 1
    • Fevipiprant, BIO-11006, losmapimod, nemiralisib, lebrilizumab are all terminated  - The authors should carefully review the therapies do determine their current status ( and then note in the table/paper when this was done- as this changes quickly)

Response: The tables have been revised and updated. We made a change in the column that indicated whether the study was discontinued or not and the column now presents the current status of the therapy. It is imperative to avoid the misguidance of the reader.

  • Remove Relovair as trademark names are different globally

Response: All trademark names have been removed as requested.

  • Some of the therapies listed are actually approved in the treatment of other diseases.  Perhaps  an additional column or some type of notation could indicate this.

Response: A column has been added to indicate the therapies that are used to treat other illnesses.

  • Table 2 studies most completed and reported

Response: As suggested, we added new studies and more information to the table.

  • May be useful to add inhaled, oral, antibody to Tables 1&2

Response: As suggested, a column has been added that describes how to administer the drugs.

  • Line 497 Smoking cessation or tobacco treatment (vs quitting smoking)

Response: The sentence that contained the information to quit smoking was rephrased for smoking cessation.
